# Short communication: Tissue distribution of major cannabinoids following intraperitoneal injection in male rats

Cody A. C. Lust[1], Xinjie Lin[1], Erin M. Rock[2], Cheryl L. Limebeer[2], Linda A. Parker[2], David W. L. Ma[1] *

1 Department of Human Health and Nutritional Sciences, University of Guelph, Guelph, ON, Canada,
2 Department of Psychology and Collaborative Neuroscience Program, University of Guelph, Guelph, ON, Canada

* davidma@uoguelph.ca

**Data Availability Statement:** All relevant data are within the manuscript and its Supporting Information files.

## Abstract

Currently, peripheral tissue distribution of cannabinoids after treatment is poorly understood. This pilot study sought to examine the early tissue distribution of major cannabinoids 30 minutes following an intraperitoneal injection of vehicle (1:9 Tween 80/SAL), and doses of THC (1 mg/kg) and CBD (5 mg/kg) that are feasible for human consumption in serum, adipose, brain, lung, liver, jejunum, and muscle of male Sprague-Dawley rats. The jejunum and adipose were most enriched in THC. Similarly, CBD was enriched in the jejunum and adipose but also the liver. In contrast, the brain had the lowest concentration of cannabinoids relative to other tissues. The liver had the greatest concentration of the THC metabolites, 11-OH-THC and COOH-THC, compared to all other tissues. Overall, these findings highlight broad tissue distribution and marked differences in tissue concentration not previously appreciated. Thus, as cannabinoid research continues to rapidly grow, consideration of the potential bioactive effects of these molecules in peripheral tissues is warranted in future studies.

## Introduction

Legalization of cannabis across Canada and select U.S. states has led to an increase of cannabis-based products being consumed recreationally and for therapeutic outcomes related to pain and inflammation [1, 2]. Upwards of 100 different cannabinoids have been isolated to-date, however, the primary psychoactive component Δ-9-tetrahydrocannabinol (THC) and the non-psychoactive cannabidiol (CBD) are the most well-studied and abundant in commercially available products [3, 4]. Although derived from the same source, THC and CBD exhibit varying pharmacokinetics depending on the mode of consumption and exert differential biological effects [5, 6]. THC is predominantly metabolized in the liver and is converted into the psychoactive metabolite, 11-hydroxy-THC (11-OH-THC), which can be further oxidized to produce the non-psychoactive metabolite, 11-Nor-9-carboxy-THC (THC-COOH) [7]. THC exerts its psychoactive effects as a partial agonist of the $CB_1$ receptor, a G-protein coupled receptor (GPCR) in the endocannabinoid system, which is primarily expressed in the central nervous system [1, 5]. CBD can act as an inverse agonist of the $CB_1$ and $CB_2$ receptor, which

**Funding:** L.A.P Canadian Institutes of Health Research (grant No.388239) and Natural Sciences and Engineering Research Council (grant No.03629). https://cihr-irsc.gc.ca/e/193.html https://www.nserc-crsng.gc.ca/index_eng.asp Funders did not play any role in the study design, data collection and analysis, decision to publish, or preparation of the manuscript.

**Competing interests:** The authors have declared that no competing interests exist.

are predominantly expressed in immune tissues and expression is induced in the central nervous system during disease or injury [3, 5]. The primary neuroprotective and additional anti-inflammatory effects of CBD are attributed to its role as an agonist of the serotonin GPCR 5-HT$_{1A}$ and the vanilloid TRPV1 receptors, found in the central nervous system [8].

While there is some understanding of differences in bioavailability due to the route of administration, there remains limited understanding of the distribution and concentration of cannabinoids in different organs and tissues. For example, following oral consumption, CBD and THC undergo significant first-pass metabolism in the liver which limits bioavailability to <10%, while inhalation, injection, and transdermal application of cannabinoids bypass the liver resulting in greater bioavailability [4, 7]. Once in circulation, it has been reported in some animal models that the highly lipophilic CBD and THC may distribute to surrounding tissues and organs such as the lungs, adipose, liver, and even across the blood brain barrier [3, 9, 10]. However, studies to date have predominantly focused on blood and brain concentrations following intraperitoneal (IP) injection of CBD and THC. A more complete understanding of tissue distribution will enhance our fundamental understanding of potential sites of biological activity for CBD, THC, and metabolites, which has relevance for understanding their therapeutic potential in human disease or toxicity. Thus, our study examined the acute tissue distribution of major cannabinoids and metabolites 30 minutes following an IP injection of feasibly attainable human doses of THC (1 mg/kg body weight [BW]) and CBD (5 mg/kg BW) in male Sprague-Dawley rats.

## Material and methods

### Animals

All animal use protocols (#3941) were approved by the Institutional Animal Care Committee at the University of Guelph, which is accredited by the Canadian Council on Animal Care. Male Sprague-Dawley rats obtained from Charles River Laboratories (St Constant, QC, Canada) were used for assessment. Rats were individually housed in opaque plastic cages (48 x 26 x 20 cm), containing bed-o-cob bedding from Harlan Laboratories, Inc. (Mississauga, ON, Canada), a brown paper towel and Crink-l'Nest™ from The Andersons, Inc. (Maumee, OH, USA) and a white paper cup that was 14 cm long and 12 cm in diameter. The colony room was maintained at an ambient temperature of 21˚C and a 12/12 h reverse light–dark schedule (lights off at 7 a.m.). Rats were maintained on ad libitum chow and water.

### Cannabinoids

Synthetic THC (98% pure; Toronto Research Chemicals) and CBD (97.4% pure; Toronto Research Chemicals) were first dissolved in ethanol in a graduated cylinder. Tween 80 (Sigma) was added to the solution, and the ethanol was evaporated off with a gentle nitrogen stream, after-which saline (SAL) was added. The final vehicle (VEH) solution consisted of 1:9 Tween 80/saline. CBD was prepared to obtain a working concentration of 5 mg/ml and administered as an IP injection at 1 ml/kg (5 mg/kg BW). THC was prepared to obtain a working concentration of 1 mg/ml and administered as an IP injection at 1 ml/kg (1 mg/kg BW). The relatively low doses of CBD and THC were chosen to reflect the upper-range of physiologically feasible human intakes of CBD and THC [11]. Six male rats were included in each of the VEH, CBD, and THC groups.

### Sample collection

Rats were sacrificed 30 minutes post-injection by rapid decapitation (restrained in a decapi-cone; Braintree Scientific, MA, USA), and tissue (brain, epididymal adipose, liver, lung, muscle

of the hind leg, blood, and the jejunum of the small intestine) was immediately collected. Then, tissues were flushed with phosphate-buffered saline, flash frozen in liquid nitrogen, and stored at -80˚C. All tissue samples were collected from the same site and whole organs were collected (where applicable).

## Extraction and quantification of metabolites by LC-MS/MS

The method to extract and quantify CBD, THC, and THC metabolites 11-OHC-THC & THC-COOH from tissues was adapted from [12] and conducted by the Analytical Facility for Bioactive Molecules (AFBM) at SickKids Hospital (Toronto, Ontario, Canada). THC-D3 (internal standard) was purchased from Sigma (St. Louis, MO, USA) and the same quantity was added to all samples. Random frozen samples, 50–80 mg, were weighed and transferred into Precellys homogenization tubes (containing ceramic beads; Bertin Technologies, Rockville, Washington DC) and $H_2O$ was added to each tube to achieve a target concentration of 100 mg/mL and homogenized using a Precellys 24 high-throughput homogenizer (Bertin Technologies). Homogenized suspension, 200 μL, (corresponding to 20 mg tissues) was transferred into siliconized round bottom tubes containing the internal standard THC-D3 (100 ng/ml) and 800 μL of $H_2O$ for a final volume of 1 mL. For serum, 50 μL of samples was added to siliconized tubes containing 950 μL of $H_2O$ instead. An eleven-point standard curve (0–2000 ng) was prepared for each individual tissue. Then, 20 μL of 1 M HCl was add to round bottom tubes and briefly vortexed. Subsequently, 2 mL of 9:1 (v/v) hexane/ethyl acetate was added followed by a brief vortex and centrifuged at 200 x g for 10 minutes. The supernatant was removed and transferred to siliconized conical glass tubes. The extraction with 2 mL of 9:1 (v/v) hexane/ethyl acetate was repeated and the supernatants were combined. Samples and standards were evaporated under a gentle flow of nitrogen with the heater block set to 35˚C. The residues were reconstituted in 120 μL of acetonitrile (ACN), centrifuged at 500 x g for 2 minutes with the remaining supernatant transferred into 200 μL glass inserts for liquid chromatography with tandem mass spectrometry (LC-MS/MS) analysis. An Agilent 1290 ultra performance liquid chromatography system (Agilent Technologies, Santa Clara, CA, USA) fitted with a Sciex Q-Trap 5500 mass spectrometer (AB Sciex, Framingham, MA, USA) was used in electron spray ionization mode. Samples were separated using a Kinetex Biphenyl column (2.6 μm, 100Å, 50 x 2.1 mm; Phenomenex, Torrence, CA, USA). A gradient mobile phase of 5 minutes at a flow rate of 0.4 ml/min was used for the elution of THC with mobile phase A (MPA): 0.1% formic acid in water and mobile phase B (MPB): 0.1% formic acid in ACN. The mobile phase gradient was: t = 0 minutes 50% B, t = 2.50 minutes 95% B, t = 2.55 minutes 50% B and t = 4.50 minutes 50%. Data was collected and analyzed by Analyst v 1.6.2 (Sciex). All LC-MS/MS grade solvents were purchased from Caledon Laboratories Ltd (Georgetown, ON, Canada).

## Data analysis

All analyses were conducted in SAS University Edition (SAS Institute Inc.; Cary, North Carolina). Cannabinoid results outside of the highest or lowest point on the standard curve were not included in the calculation of the mean and standard deviation for their respective tissues.

## Results

Table 1 reports concentrations of 4 major cannabinoids in 7 different tissues. No cannabinoids were found in any of the tissues collected from rats that received VEH (data not shown). The jejunum and adipose were observed to have the highest concentrations of THC 30 minutes following IP injection. CBD was also enriched in the jejunum and adipose, but also the liver.

**Table 1. Concentration of cannabinoids 30 minutes following IP injection.**

| CANNABINOID | ADIPOSE | BRAIN | JEJUNUM | LIVER | LUNG | MUSCLE | SERUM |
|---|---|---|---|---|---|---|---|
| THC[a] | 2947.5 ± 705.9 | 28.9 ± 8.5 | 2258.3 ± 1367.9 | 55.2 ± 41.1 | 49.1 ± 24.5 | 190.0 ± 387.4 | 18.1 ± 5.6 |
| 11-OH-THC[a] | n/c | 72.9 ± 38.2 | 110.15 ± 69.9[c] | 414.8 ± 128.5 | 87.3 ± 29.0[c] | 55.2 ± 20.8 | 25.5 ± 13.4 |
| 11-COOH-THC[a] | 7.6 ± 1.7[c] | n/d | 40.6 ± 14.8 | 307.2 ± 99.5 | **n/c** | n/d | 22.5 ± 16.2 |
| CBD[b] | 1013.3 ± 428.3[c] | 153.8 ± 52.2 | 10372.0 ± 11351.4[d] | 2810.0 ± 1133.5 | **465.3 ± 296.3** | 176.0 ± 77.5 | 118.1 ± 32.9 |

Values are expressed as ng/g of tissue. Serum is expressed as ng/ml ± standard deviation. n/d; Non-detectable peak. n/c; Not enough data points to calculate. n = 6 for all groups unless otherwise specified.

[a]Only rats injected with THC were included in analysis.

[b]Only rats injected with CBD were included in analysis.

[c]n = 3; 3 samples were excluded which were outside the highest or lowest point of the standard curve.

[d]n = 4; 2 samples were excluded which were outside the highest or lowest point of the standard curve.

Brain had the lowest concentration of cannabinoids relative to other tissues. In rats which received THC injections, the liver had the highest concentration of THC metabolites, 11-OH-THC and COOH-THC, when compared to all other tissues. Serum was the only other tissue which had greater concentrations of 11-OH-THC and COOH-THC compared to concentrations of THC. CBD was given at a concentration of 5 mg/kg compared to 1 mg/kg of THC, thus as expected, higher concentrations of CBD was found comparatively to THC in a majority of tissues. All relevant data are within the manuscript and S1 Data.

## Discussion

To the authors' knowledge, this is one of the first studies conducted which has described the concentration of multiple cannabinoids in 7 different tissues. This pilot study provides an important whole-body perspective on the potential distribution and therefore potential sites of action. This is important given that cannabinoid activity and metabolism across tissues are likely interrelated. Low doses of both CBD and THC were chosen to reflect values feasible for human consumption. Herein, we report that the jejunum and adipose were enriched by either THC or CBD (Table 1), which is likely due to the close proximity to the injection site [5]. Our results also show the highest concentration of THC metabolites, 11-OH-THC and COOH-THC, were found in the liver. The liver is the main site of THC metabolization, thus it is expected that the highest concentration of metabolites are found in this tissue, which is consistent with known pharmacokinetic studies of THC metabolites in rodents [7].

Brain and blood (whole, serum and plasma) are the most commonly assessed tissues reported in the literature. The mean concentration of 28.9 ng/g and 18.1 ng/ml of THC found in the brain and serum respectively, are similar in range with other studies reporting concentrations following IP injection in rats ranging from 0.5–10 mg/kg of THC 30 minutes to 2 hours post-injection [6, 9, 13, 14]. Similar concentrations of CBD, as reported in Table 1, have also been seen in other rodent models following IP injection ranging from 5–10 mg/kg of CBD 30 minutes to 1-hour post-injection [6, 15, 16]. With the relatively short timeframe of our tissue collection following IP injection, 30 minutes, it is notable that detectable levels of both CBD and THC were found in the brain. CBD has exhibited potent neuroprotective properties via its actions on 5-$HT_{1A}$ which is predominantly found in the brain and spleen [8]. $CB_1$ receptors are also primarily located in the central nervous system but, growing evidence shows $CB_1$ has a wider distribution outside of the central nervous system in peripheral tissues, much like the $CB_2$ receptor [17]. Thus, CBD and THC could impart biological actions, including

neuroprotective and analgesic effects, by acting on both centrally and peripherally located receptors in a relatively short amount of time [3, 8].

## Limitations

This was a pilot study and several limitations are noted for future work. MS quantification is inherently variable [18], thus, data from this study may be used to estimate sample size requirements in future studies. Future work would benefit from examining tissues at multiple timepoints, allowing for a better assessment of cannabinoid kinetics and distribution over time. We have shown that cannabinoids distribute widely in whole tissues, but there may be potential differences within specific tissue regions that have distinct biological roles. Although IP injection is commonly used in experimental studies, oral intake and inhalation are far more common and relevant methods of consumption in real world scenarios which markedly affects absorption and metabolism. Our study only included male rats, thus potential sex differences due to metabolic differences is unknown and should be investigated in future work [13].

## Conclusion

The findings of the present study enhance our understanding of the tissue distribution of cannabinoids. By examining whole body tissue distribution this pilot study sheds light on the potential activity of THC and CBD on a variety of tissues which may help to inform future study designs examining the broader biological effects of cannabinoids. This study demonstrated wide variation in the tissue distribution and concentrations of major cannabinoids in 7 different tissues following IP injection of THC or CBD in rats. Thus, consideration of peripherical tissues in future cannabinoid research is warranted.

## Supporting information

**S1 Data.**
(XLSX)

## Acknowledgments

We would like to thank Ashley St. Pierre and Fatima Sultani from AFBM at SickKids Hospital, Toronto, Canada for assistance with the cannabinoid analysis.

## Author Contributions

**Conceptualization:** Cody A. C. Lust, Linda A. Parker, David W. L. Ma.

**Formal analysis:** Cody A. C. Lust, Cheryl L. Limebeer.

**Funding acquisition:** Linda A. Parker, David W. L. Ma.

**Methodology:** Xinjie Lin, Cheryl L. Limebeer.

**Project administration:** Linda A. Parker, David W. L. Ma.

**Supervision:** David W. L. Ma.

**Writing – original draft:** Cody A. C. Lust, David W. L. Ma.

**Writing – review & editing:** Cody A. C. Lust, Xinjie Lin, Erin M. Rock, Cheryl L. Limebeer, Linda A. Parker, David W. L. Ma.

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
