## [Decision Letter · Decision Letter 0]

2 Nov 2021

PONE-D-21-30787Short Communication:Tissue distribution of major cannabinoids following intraperitoneal injection in male ratsPLOS ONE

Dear Dr. Ma,

Thank you for submitting your manuscript to PLOS ONE. After careful consideration, we feel that it has merit but does not fully meet PLOS ONE’s publication criteria as it currently stands. Therefore, we invite you to submit a revised version of the manuscript that addresses the points raised during the review process. I have now received the comments of two experts in the field. They both find your manuscript interesting that addresses an important issue. Nevertheless, there are points of criticism asking for further clarification.

One aspect is in my eyes of special importance. Looking at the standard deviations of values in table 1 the question raises how reliable the results might be for other reserchers referencing to this work. 

In part there are deviations of more than 200%. Please raise the number of animals and include female rats to the study.

Furthermore, the area of tissue collection should clearly be specified as different brain areas, adipse tissue (subcutanous or visceral) and striated muscles might strongly affects the results.

We look forward to receiving your revised manuscript.

Kind regards,

Faramarz Dehghani

Academic Editor

PLOS ONE

Journal Requirements:

"Funding provided to L.A.P from the Canadian Institutes of Health Research (grant No.388239) and National Sciences and Engineering Research Council (grant No.03629). We would like to thank Ashley St. Pierre and Fatima Sultani from AFBM at SickKids Hospital, Toronto, Canada for assistance with the cannabinoid analysis."

"L.A.P

Canadian Institutes of Health Research (grant No.388239) and National Sciences and Engineering Research Council (grant No.03629).

https://cihr-irsc.gc.ca/e/193.html

https://www.nserc-crsng.gc.ca/index_eng.asp

Funders did not play any role in in the study design, data collection and analysis, decision to publish, or preparation of the manuscript"

Reviewers' comments:

Reviewer's Responses to Questions

**Comments to the Author**

1. Is the manuscript technically sound, and do the data support the conclusions?

Reviewer #1: Partly

Reviewer #2: Partly

2. Has the statistical analysis been performed appropriately and rigorously? 

Reviewer #1: I Don't Know

Reviewer #2: Yes

3. Have the authors made all data underlying the findings in their manuscript fully available?

Reviewer #1: Yes

Reviewer #2: Yes

4. Is the manuscript presented in an intelligible fashion and written in standard English?

Reviewer #1: Yes

Reviewer #2: Yes

5. Review Comments to the Author

Reviewer #1: The scientific question in the article titled “Tissue distribution of major cannabinoids in male rats” is an important contribution to the field. As a short communication, it meets the general standard of a simple, straight-forward set of data that stand alone and meet a specific scientific need. There are a few clarification points for methodology that must be addressed to provide further clarification and justification.

1) Why did the authors use two different doses of THC (1mg/kg) and CBD (5mg/kg)? It is curious that the authors did not use the same dose, nor do they justify the use of different doses, yet they compare the doses as if they are the same. This must be addressed in the methods and the discussion.

2) The areas of evaluation (brain, gut, liver, adipose, muscle) are extremely large and varied; however, the methods appear to state that 20mg of tissue were used for analysis (that size being understandable). The authors should provide very accurate details on which areas of these organs were sampled and that this sampling was consistent across subjects. Simply sampling in different areas of these organs could cause variability.

3) The authors should list the recovery rate for the internal standard for each tissue type and state how this was adjusted across analyses. If the internal standard recovery was not used as an adjustment factor across tissue, they should explain why and justify this level of analysis.

Reviewer #2: The authors used rats to measure the distribution of THC and CBD in several tissues 30 min after intraperitoneal injection. Even if it is generally an interesting topic, the results are of relatively little informative value. As already mentioned in the limitations: the sample size is low, there is only one time point, sex differences are unknown. Moreover, more complete pharmacokinetic studies in rats regarding THC and CBD have already been published. What is new here is the measurement of cannabinoids in additional tissues. But especially for these tissues, a time course is missing.

There is only one point that I would like to be addressed by the authors: Please add to the limitations that the manner of administration (intraperitoneal injection) of the cannabinoids does not reflect the common way of administration. Usually, an oral (CBD) or pulmonary (THC) administration should be preferred as it was performed e.g. by Hlozek et al. 2017 (your reference 6).

6. PLOS authors have the option to publish the peer review history of their article (what does this mean?). If published, this will include your full peer review and any attached files.

Reviewer #1: **Yes: **Heather B Bradshaw

Reviewer #2: No

---

## [Author Response · Author response to Decision Letter 0]

16 Dec 2021

Journal Requirements:

Style requirements have been revised throughout.

Funding related statements have been removed from the acknowledgements section of the manuscript. The funding statement previously provided online is correct. 

Academic Editor:

1) Looking at the standard deviations of values in table 1 the question raises how reliable the results might be for other researchers referencing to this work. 

We acknowledge in the limitations section of the manuscript values with higher standard deviations, but also highlight that these are more realistic values than typical SEM values reported in the literature. Thus, our goal was to provide a resource to compare and contrast a large number of tissues and a quantitative ‘estimate’ for what may be detectable, providing new directions for other researchers to build upon these ideas in the future. This provides a springboard for further investigation as we acknowledge that values may further differ depending upon route of exposure, duration and timing of measurements. Nevertheless, values of the brain and serum, described in Table 1, are similar to those of other papers that have been previously published (lines 178-183) giving us confidence in the methodology used in our paper. 

2) In part there are deviations of more than 200%. Please raise the number of animals and include female rats to the study.

We agree these are meritorious recommendations and should certainly be considered in future work. As a pilot study in a relatively new field of research, this focused work highlights important conceptual ideas and a starting point for future examinations. Importantly, it enables us to determine what is an appropriate sample size/power calculation and other aspects we can build upon for future work. This perspective is encapsulated by comments of reviewer 1 highlighting the scope and intention of our work: “As a short communication, it meets the general standard of a simple, straight-forward set of data that stand alone and meet a specific scientific need.”

3) Furthermore, the area of tissue collection should clearly be specified as different brain areas, adipose tissue (subcutaneous or visceral) and striated muscles might strongly affects the results.

We have addressed a similar comment #2 from Reviewer 1. See below for our revisions.

Reviewer #1: 

The scientific question in the article titled “Tissue distribution of major cannabinoids in male rats” is an important contribution to the field. As a short communication, it meets the general standard of a simple, straight-forward set of data that stand alone and meet a specific scientific need. There are a few clarification points for methodology that must be addressed to provide further clarification and justification.

1) Why did the authors use two different doses of THC (1mg/kg) and CBD (5mg/kg)? It is curious that the authors did not use the same dose, nor do they justify the use of different doses, yet they compare the doses as if they are the same. This must be addressed in the methods and the discussion.

This is a very astute observation and one we agree requires further clarification. We utilized relatively low doses to better reflect what would be a physiologically relevant intake of cannabinoids. A dose of 5 mg/kg ip CBD is a relatively low dose employed in the field, yet a dose of 5 mg/kg THC is a relatively high dose—1 mg/kg THC is a more appropriate comparison given the behavioral effects of these doses. Rodent models often utilize quite a high dose in their methodology (eg. 10+ mg/kg THC and CBD) which is far beyond what any human would ingest. That would be upwards of 860mg of THC or CBD for the average adult male. As THC elicits psychoactive effects, it is not tolerated as well at higher doses compared to CBD thus, the discrepancy between doses. We have updated the methods section (lines 103-105) to highlight this fact. It was not our intent to compare THC and CBD and the text has been revised throughout.

2) The areas of evaluation (brain, gut, liver, adipose, muscle) are extremely large and varied; however, the methods appear to state that 20mg of tissue were used for analysis (that size being understandable). The authors should provide very accurate details on which areas of these organs were sampled and that this sampling was consistent across subjects. Simply sampling in different areas of these organs could cause variability.

This is a great observation as we agree there can be potential variation within areas of an organ/tissue (eg. lobes of the liver). This was not specifically controlled and have included this as a limitation in our analyses (as described in lines 196-197). 

3) The authors should list the recovery rate for the internal standard for each tissue type and state how this was adjusted across analyses. If the internal standard recovery was not used as an adjustment factor across tissue, they should explain why and justify this level of analysis.

In the methodology, a known amount of internal standard (THC-D3) was added to all the samples for the purpose of quantification and suitable approach to obtain a first estimate. As this was a pilot study, the internal standard recovery rate was not assessed. This limitation is will be noted for future quantitative work.

Reviewer #2: 

The authors used rats to measure the distribution of THC and CBD in several tissues 30 min after intraperitoneal injection. Even if it is generally an interesting topic, the results are of relatively little informative value. As already mentioned in the limitations: the sample size is low, there is only one time point, sex differences are unknown. Moreover, more complete pharmacokinetic studies in rats regarding THC and CBD have already been published. What is new here is the measurement of cannabinoids in additional tissues. But especially for these tissues, a time course is missing.

1) Please add to the limitations that the manner of administration (intraperitoneal injection) of the cannabinoids does not reflect the common way of administration. Usually, an oral (CBD) or pulmonary (THC) administration should be preferred as it was performed e.g. by Hlozek et al. 2017 (your reference 6).

Thank you for your feedback highlighting the novelty of work, which provides new information in additional tissues. This work sheds light on the potential impact of CBD and THC in other tissues that have not been previously considered. We agree that i.p. injection is certainly not the most common mode of consumption in a real world setting. However, a large number of preclinical studies have used i.p. injection as their method of administration and our goal was to use the most common approach in literature to increase the comparability of our results with past work. Lines 198-201 have been added in our limitations to reflect this idea.

---

## [Decision Letter · Decision Letter 1]

3 Jan 2022

Short Communication:Tissue distribution of major cannabinoids following intraperitoneal injection in male rats

PONE-D-21-30787R1

Dear Dr. Ma,

We’re pleased to inform you that your manuscript has been judged scientifically suitable for publication and will be formally accepted for publication once it meets all outstanding technical requirements.

Kind regards,

Faramarz Dehghani

Academic Editor

PLOS ONE

Additional Editor Comments (optional):

Reviewers' comments:

Reviewer's Responses to Questions

**Comments to the Author**

1. If the authors have adequately addressed your comments raised in a previous round of review and you feel that this manuscript is now acceptable for publication, you may indicate that here to bypass the “Comments to the Author” section, enter your conflict of interest statement in the “Confidential to Editor” section, and submit your "Accept" recommendation.

Reviewer #2: All comments have been addressed

2. Is the manuscript technically sound, and do the data support the conclusions?

Reviewer #2: Yes

3. Has the statistical analysis been performed appropriately and rigorously? 

Reviewer #2: Yes

4. Have the authors made all data underlying the findings in their manuscript fully available?

Reviewer #2: Yes

5. Is the manuscript presented in an intelligible fashion and written in standard English?

Reviewer #2: Yes

6. Review Comments to the Author

Reviewer #2: The authors have addressed all my comments. They have extended the limitations following my suggestion. I don't have any further concerns.

7. PLOS authors have the option to publish the peer review history of their article (what does this mean?). If published, this will include your full peer review and any attached files.

Reviewer #2: No

---

## [Editor Report · Acceptance letter]

10 Jan 2022

PONE-D-21-30787R1 

Short Communication: Tissue distribution of major cannabinoids following intraperitoneal injection in male rats 

Dear Dr. Ma:

I'm pleased to inform you that your manuscript has been deemed suitable for publication in PLOS ONE. Congratulations! Your manuscript is now with our production department. 

Kind regards, 

on behalf of

Dr. Faramarz Dehghani 

Academic Editor

PLOS ONE